# Structural pharmacology of SV2A reveals an allosteric modulation mechanism in the major facilitator superfamily

Shabareesh Pidathala [1], Xiao Chen[1], Yaxin Dai [1], Long N. Nguyen [2], Christoph Gorgulla [1], Yiming Niu[3], Fangyu Liu [4] ✉ & Chia-Hsueh Lee [1] ✉

The synaptic vesicle glycoprotein 2A (SV2A), a member of the major facilitator superfamily (MFS), is a key target for antiseizure medications and a biomarker for synaptic density imaging. Despite its clinical importance, the mechanisms underlying SV2A ligand binding and modulation remain poorly understood. Here, we report sub-3 Å resolution cryo-electron microscopy (cryo-EM) structures of human SV2A in its apo form and in complex with FDA-approved antiseizure medication levetiracetam; PET imaging tracer UCB-J; experimental antiseizure drug padsevonil; and allosteric modulator UCB1244283. We find that levetiracetam and UCB-J induce vestibule occlusion, a hallmark conformational transition of MFS transporters that had not been observed in previous SV2A structures. UCB1244283 binds to an allosteric site and enhances orthosteric ligand engagement by stabilizing the occluded state and slowing ligand dissociation. Notably, padsevonil occupies both orthosteric and allosteric sites, functionally precluding modulation. These findings uncover an allosteric mechanism of regulation and provide a structural framework for the development of modulators targeting SV2A and related MFS transporters.

Solute carrier (SLC) transporters constitute the second largest family of human membrane proteins, after G-protein coupled receptors[1,2]. With nearly 400 members grouped into 65 different families, SLCs transport a variety of solutes across lipid bilayers[1,2] and are pharmacological targets for a variety of antidiabetic, neuropsychiatric, and diuretic drugs[3]. Besides conventional drug design, development of allosteric modulators targeting SLC transporters is being actively pursued[4,5]. Allosteric ligands typically occupy distinct sites along the vestibules of these transporters and modulate ligand binding at the orthosteric site[5]. Among SLCs, structural characterization of allosteric modulation has been reported for transporters with LeuT fold (e.g., leucine transporter (LeuT), serotonin transporter (SERT), dopamine transporter (DAT)) or GltPh fold (e.g., excitatory amino acid transporter (EAAT))[6–10]. On the other hand, major facilitator superfamily (MFS) transporters are the largest group within SLCs[1], but the

identification and characterization of allosteric sites in MFS members remain limited.

In this study, we present the structural and functional mechanisms of allosteric modulation in a putative MFS transporter, synaptic vesicle glycoprotein 2A (SV2A). Along with its related isoforms SV2B and SV2C, SV2A is a member of the SLC22B subfamily. SV2A is expressed in nearly all neurons, and mass spectrometry studies show that it is one of the most abundant membrane proteins in synaptic vesicles, with 5–12 copies present in each vesicle[11–13], indicating a function relevant to neurotransmission. Due to its ubiquitous presence and high expression throughout the brain, SV2A serves as a marker for non-invasive neuroimaging of synaptic density[14]. For instance, the SV2A ligand UCB-J has been used as a positron emission tomography (PET) tracer to study Alzheimer's and Parkinson's diseases[15,16].

---

[1]Department of Structural Biology, St. Jude Children's Research Hospital, Memphis, TN, USA. [2]Department of Biochemistry, Yong Loo Lin School of Medicine, National University of Singapore, Singapore, Singapore. [3]Laboratory of Chromosome and Cell Biology, The Rockefeller University, New York, NY, USA. [4]Department of Pharmacology, UT Southwestern Medical Center, Dallas, TX, USA. ✉e-mail: fangyu.liu@utsouthwestern.edu; chiahsueh.Lee@stjude.org

SV2A is also known to play a role in epilepsy[17]. Genetic knockout of SV2A in mice results in severe seizures and death within the first few weeks of life[18]. Mutations in human SV2A lead to intractable epilepsy, developmental delays, and cognitive impairment[19]. Although its endogenous substrate and transport function remain elusive, SV2A is a target for clinical antiseizure drugs such as levetiracetam and brivaracetam. Padsevonil, a compound with higher affinity for SV2A compared to levetiracetam and brivaracetam, was previously tested in clinical trials for drug-resistant epilepsy[20,21]. UCB1244283 is a compound that selectively enhances the binding of levetiracetam and brivaracetam, but intriguingly, it does not affect padsevonil[20,22]. These observations highlight the complexity of SV2A's interactions with therapeutic compounds. Recent structural studies on SV2A have provided insights into the binding of racetam-based ligands levetiracetam[23,24], brivaracetam[23,25], and UCB2500[26]. However, the precise mechanisms of underlying drug action remain unclear, as those studies indicated that levetiracetam and brivaracetam binding did not induce conformational changes in SV2A[23,24]. Moreover, structural discrepancies exist across studies, in some cases arising from atypical oligomerization states[23,25]. Finally, the basis of allosteric modulation by UCB1244283 is still entirely unknown.

Here we report cryo-EM structures of human SV2A in its ligand-free (apo) state and in complex with levetiracetam, UCB-J, UCB-J/UCB1244283, levetiracetam/UCB1244283, and padsevonil, along with functional studies. We find that levetiracetam and UCB-J bind to the central cavity and induce the occlusion of SV2A, a mechanism of action that had not been observed previously. UCB1244283 binds to a secondary ligand-binding site in SV2A, and this pocket acts as an allosteric site that potentiates binding of ligands to the primary, orthosteric site. We also show that padsevonil binds to both the primary and secondary sites, thus precluding the action of UCB1244283. Our study offers a rare view of a bona fide allosteric site within MFS transporters and provides a foundation for developing therapeutic ligands targeting the allosteric site of SV2A.

## Results

### SV2A structure in the absence of ligand

To facilitate cryo-EM studies of SV2A, we used a fusion strategy that previously allowed us to elucidate high-resolution structures of other challenging MFS transporters[27–29]. We replaced 140 residues from the predicted disordered N-terminus of SV2A with a maltose-binding protein (MBP) tag ($SV2A_{EM}$) (Supplementary Fig. 1A). We tested the function of $SV2A_{EM}$ by measuring the competition between unlabeled compounds and $^3$H-UCB-J. $SV2A_{EM}$ exhibited binding responses to levetiracetam, UCB-J, and padsevonil comparable to those of wild-type SV2A, indicating that the construct modification does not impair drug binding (Fig. 1A). We reconstituted purified $SV2A_{EM}$ protein into saposin nanoparticles (Supplementary Fig. 1A) and formed complexes with an MBP-specific DARPin[30] to further increase the particle size for cryo-EM (Supplementary Fig. 1B).

We determined the structure of $SV2A_{EM}$ in the absence of ligands at 2.8 Å resolution (Supplementary Fig. 2A). The transmembrane core (Fig. 1B) and three short intracellular helices (ICH0−2; Fig. 1C) were well resolved. In contrast, the luminal domain was discernible in the 2D classification and unsharpened map (Supplementary Fig. 3A), but lacked high-resolution features, likely due to its intrinsic flexibility. To ensure that our construct modification did not compromise the structural integrity of this domain, we also determined the structure of $SV2A_{EM}$ in complex with the receptor-binding domain of BoNT/A1 (Supplementary Fig. 3B), a Clostridium toxin known to bind the luminal domain of SV2A[31]. Our results show that $SV2A_{EM}$ can bind BoNT/A1, and the binding mode is identical to that previously reported

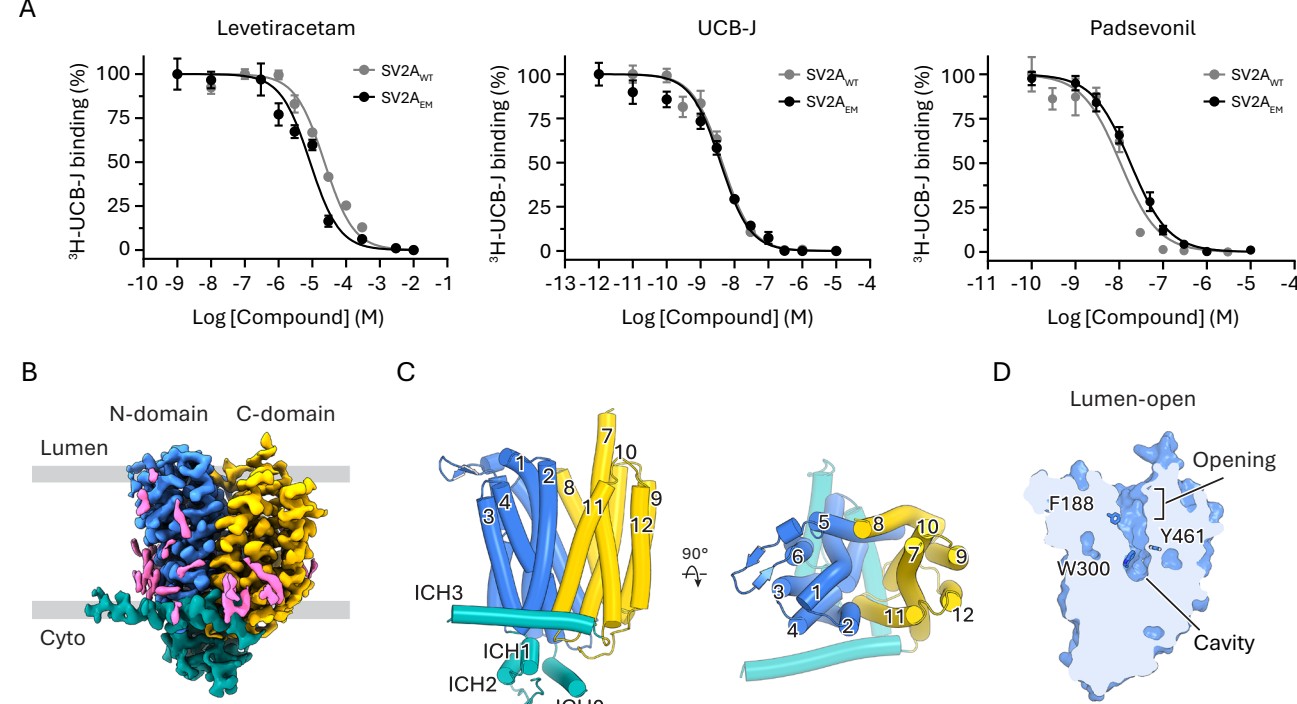

**Fig. 1 | Functional characterization and cryo-EM structure of human SV2A in apo state. A** Dose-response curves for competitive displacement of 10 nM $^3$H-UCB-J bound to either $SV2A_{WT}$ or $SV2A_{EM}$ containing membranes. Data shown as mean ± s.d.; n = 6 biological replicates. For each ligand, 100% and 0% binding were defined as $^3$H-UCB-J signal obtained at the lowest and highest concentration of ligand used, respectively. **B** Cryo-EM density of SV2A in a ligand-free, apo state.

Lipid-like densities are in magenta. **C** MFS-fold architecture of SV2A with the N-terminal (blue) 6 TMs related to the C-terminal (yellow) 6 TMs by pseudo two-fold symmetry. Four intracellular helices (ICH0 to ICH3, green) lie parallel to the membrane. **D** Cut-open view of SV2A showing its lumen-open conformation, with the central cavity accessible to the lumen side of the vesicle. Source data for (**A**) are provided as a Source data file.

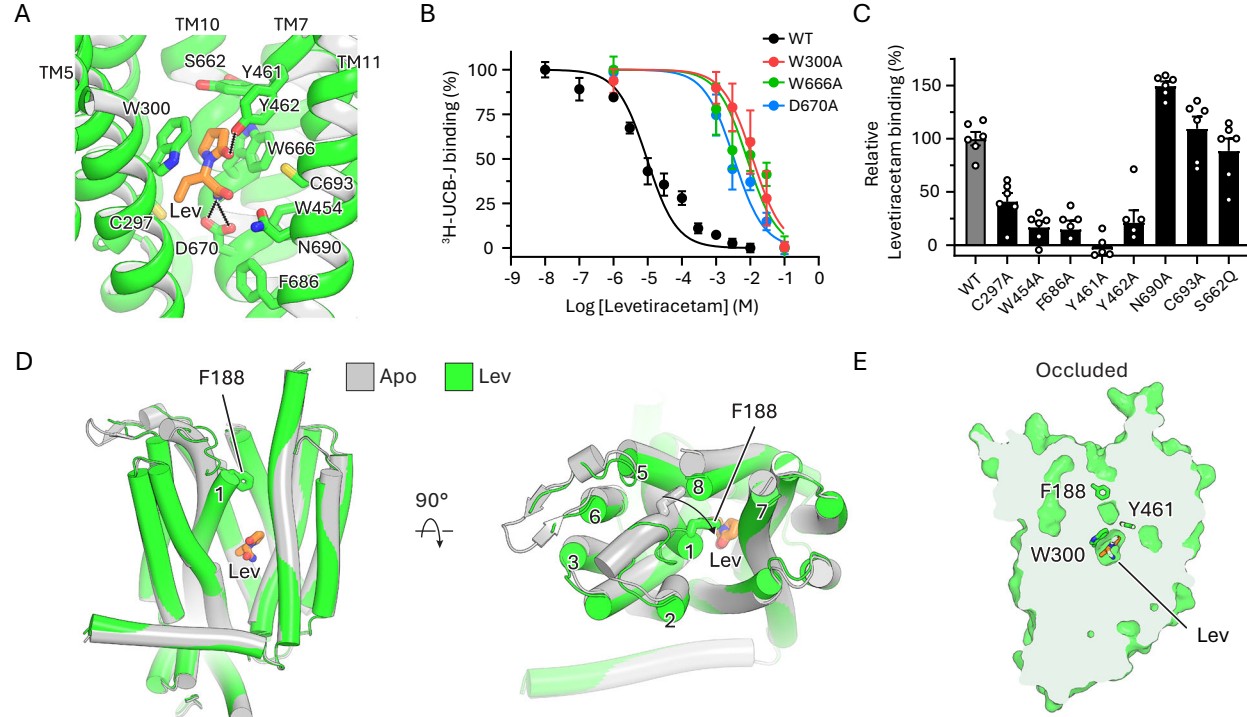

**Fig. 2 | Structure of human SV2A bound to antiseizure medication levetiracetam. A** Structure of levetiracetam (Lev, orange) bound to the central cavity of SV2A showing detailed interactions with surrounding amino acids. **B** Effect of binding site mutations on competitive displacement of 10 nM $^3$H-UCB-J by levetiracetam. Data are shown as mean ± s.d.; n = 6 biological replicates. For each binding curve, 100% and 0% were defined as $^3$H-UCB-J signal observed with the lowest and highest concentration of levetiracetam used, respectively. **C** Relative binding of levetiracetam, measured as binding of $^3$H-UCB-J (10 nM) to SV2A in the presence of 10 μM levetiracetam; binding to SV2A$_{WT}$ was defined as 100%. Data shown as mean ± s.d.; n = 6 biological replicates. **D** Superposition of SV2A-apo (gray) and SV2A–Lev (green), showing helix-to-loop transition and inward movement of TM1 in levetiracetam-bound state. **E** Cut-open view of SV2A showing that levetiracetam induces an occluded conformation.

(Supplementary Fig. 3C)[23], indicating that the luminal domain of our construct retains its native conformation. The transmembrane core adopts the canonical MFS fold, consisting of 12 transmembrane (TM) helices organized into two six-helix bundles, referred to as the N-domain and C-domain (Fig. 1B, C), connected by an intracellular segment that forms a long helix (ICH3). A central cavity lies at the interface between the N- and C-domains and is exposed to the luminal side of the synaptic vesicle, indicating that SV2A adopts a lumen-open conformation in the apo state (Fig. 1D), consistent with previous reports[23,24].

### Mechanism of levetiracetam recognition and action

To understand the mechanism of action of levetiracetam, we determined the structure of SV2A in the presence of levetiracetam at 2.7 Å resolution (Supplementary Figs. 2B and 4). The structure shows that levetiracetam occupies the central cavity of SV2A, lined by Cys297, Trp300, Trp454, Tyr461, Tyr462, Trp666, Asp670 and Phe686 (Fig. 2A). The 2-oxopyrrolidine ring of levetiracetam is sandwiched between the indole of Trp300 and Trp666, while its terminal amide interacts with the Asp670 side chain (Fig. 2A). Mutating these three key residues reduces levetiracetam binding by nearly 1000-fold (Fig. 2B). Mutations of other cavity-lining residues, such as Tyr461 and Try462, also impair levetiracetam binding (Fig. 2C), consistent with our structural observations and previous findings[23,32–34]. Phe686 and Asn690 are located further from levetiracetam but have side chains oriented toward the binding site; mutating those residues also affects drug binding, and N690A unexpectedly enhances levetiracetam binding. It is possible that these two residues contribute to the integrity of the ligand-binding pocket or to the conformational transitions of SV2A, thereby indirectly influencing ligand binding.

Comparing the structures of SV2A in its apo and levetiracetam-bound states revealed a large inward movement of TM1, along with a helix-to-loop transition around Phe188 (Fig. 2D). These structural transitions were not observed in previous SV2A–levetiracetam structures (Supplementary Fig. 5)[23,24]. The inward movement of TM1 collapses the luminal opening, with Phe188 sealing the pathway between the central cavity and the luminal side. As a result, SV2A goes from a lumen-open to an occluded conformation upon levetiracetam binding (Fig. 2E). Our findings suggest that levetiracetam exerts its pharmacological effect not only by competing with potential substrates in the central cavity but also by stabilizing the occluded state of SV2A—a previously unrecognized mechanism.

### Mechanism of UCB-J recognition and action

The SV2A-specific PET tracer UCB-J binds to SV2A$_{EM}$ with high affinity in a radioligand displacement assay, exhibiting an IC$_{50}$ of 4.1 ± 0.4 nM, 1000-fold lower than that of levetiracetam (IC$_{50}$ = 9.5 μM) (Fig. 1A). To understand the mechanisms underlying these properties, we determined the structure of SV2A in the presence of UCB-J at 2.4 Å resolution (Supplementary Figs. 2C and 4). UCB-J also occupies the central cavity of SV2A, with its core 2-oxopyrrolidine ring interacting with Trp300, Trp666, and Asp670, the same residues involved in levetiracetam binding (Fig. 3A). In addition, the trifluoro phenyl moiety of UCB-J protrudes into a groove between TM helices 7, 8 and 10, lined by the side chains of Tyr461, Val608, Ser662, Ile663 and main chain of Gly659 (Fig. 3A). The 3-methyl pyridine moiety fits snugly into a hydrophobic cavity lined by the side chains of Ile273, Trp454 and Trp666, and its ring nitrogen potentially forms a hydrogen bond with the carboxyl side chain of Asp670 (Fig. 3A). These additional interactions can account for the stronger binding of UCB-J to SV2A relative to

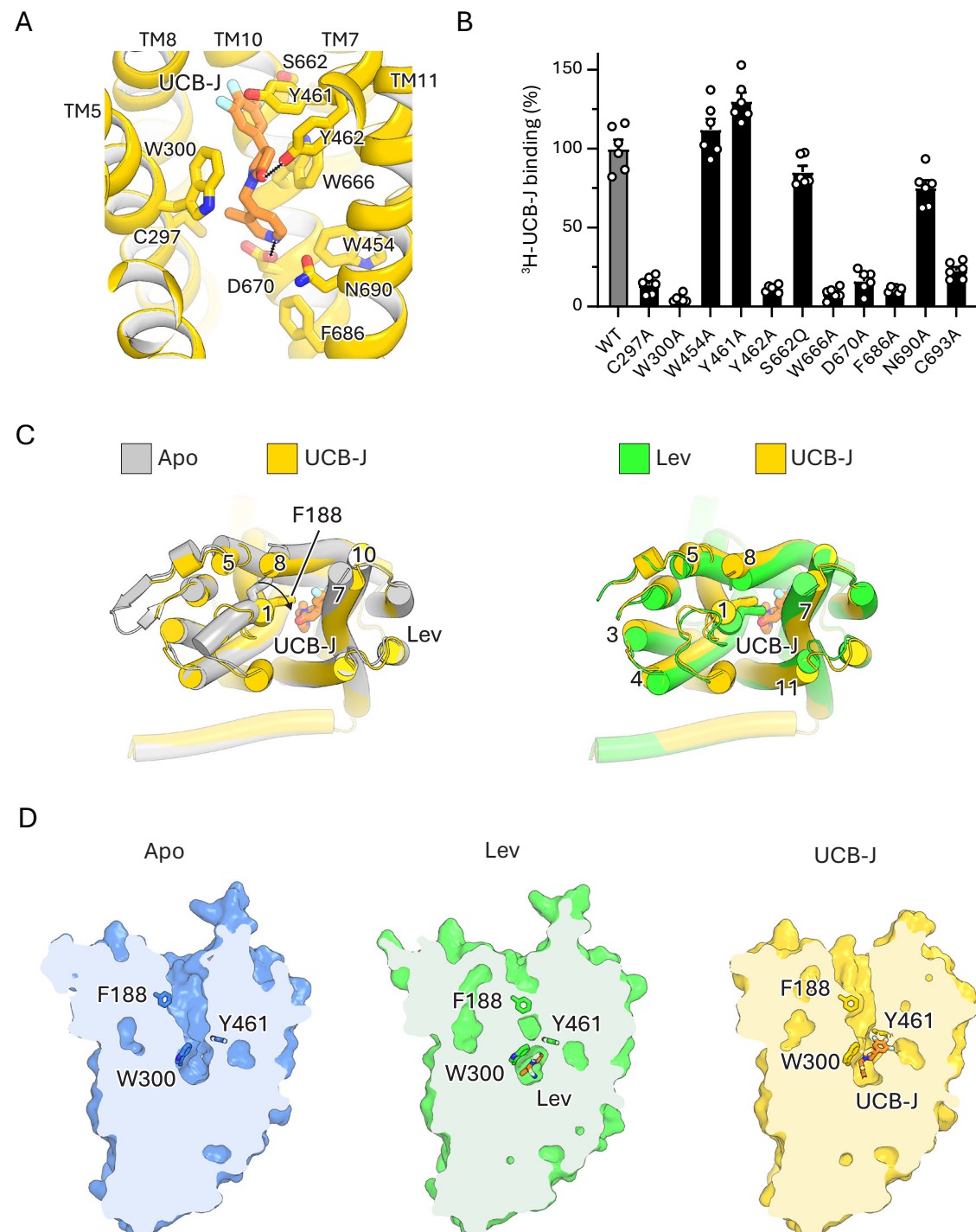

**Fig. 3 | Structure of SV2A bound to PET tracer UCB-J. A** Structure of UCB-J (orange) bound to the central site of SV2A (yellow) showing interactions with surrounding residues. **B** Effect of mutations on ³H-UCB-J (10 nM) binding. ³H-UCB-J binding to wild-type SV2A was defined as 100%. Data are shown as mean ± s.d.; n = 6 biological replicates. **C** Superposition of SV2A-apo (gray) with SV2A–UCB-J (yellow) showing the inward displacement of TM1. Overlay of SV2A–lev (green) and SV2A–UCB-J (yellow) complexes. **D** Cut-open view of SV2A showing ligand-induced occlusion of the central cavity: complete occlusion by levetiracetam (green) and partial occlusion by UCB-J (yellow). Source data for (**B**) are provided as a Source data file.

levetiracetam (Fig. 1A). Supporting our structural findings, radioligand binding experiments show that UCB-J binding is sensitive to alanine substitution of Cys297, Trp300, Tyr462, Trp666, Asp670, Phe686, or Cys693 (Fig. 3B). Mutations on Trp454 and Tyr461 do not seem to reduce UCB-J binding, in contrast to their marked effect on levetiracetam binding. Compared to levetiracetam, UCB-J is bulkier and interacts with a larger surface area of the transporter; thus, it is

possible that additional contacts help maintain UCB-J strong binding to the W454A or Y461A SV2A mutants. It is also possible that certain mutations influence different conformational transitions of the transporter, which in turn would differentially affect the binding of UCB-J and levetiracetam.

Comparison of the SV2A–UCB-J structure with the apo state reveals a substantial displacement of TM1 in the UCB-J complex

(Fig. 3C), similar to, but less pronounced than the shift observed upon levetiracetam binding (Fig. 3C). This movement constricts the central cavity, resulting in a partially occluded conformation (Fig. 3D), unlike the complete occlusion seen in the levetiracetam-bound structure. The bulkier trifluoro phenyl moiety of UCB-J appears to hinder the full inward movement of TM1, preventing Phe188 from completely covering the luminal opening (Fig. 3D). These results show that UCB-J and levetiracetam share a common mechanism, by competing for the central cavity and inducing a conformational shift in SV2A that leads to its partial or complete occlusion, respectively.

## Mechanism of allosteric modulation

UCB1244283 was previously identified as an allosteric potentiator of SV2A binding to certain drugs, including brivaracetam and levetiracetam[35]. We tested the effect of UCB1244283 on $^3$H-UCB-J binding to SV2A and observed a decrease in $K_d$ (from 19.9 to 11.1 nM), while the binding capacity increased (from 43.8 to 126 fmol/assay) (Fig. 4A).

To gain structural insights into this potentiation effect, we determined the structure of SV2A bound to both UCB-J and UCB1244283 at 2.7 Å resolution (Supplementary Fig. 2D). The transporter adopts an occluded conformation, and we observed strong density for both ligands (Supplementary Fig. 4), with UCB-J bound to the orthosteric site, as we previously observed (Fig. 3). UCB1244283 binds to a previously uncharacterized allosteric site, located approximately 13 Å above the site occupied by UCB-J (Fig. 4B). Binding of UCB1244283 to this allosteric site is stabilized by interactions with residues from TM helices 1, 5, 7, 8 and 10. The 2-methoxyaniline moiety of UCB1244283 stacks with the side chains of Phe184 (TM1) and Trp300 (TM5), while its methoxy and imine moieties form H-bonds with the side chain hydroxyl of Ser601 (Fig. 4C). The dimethylphenyl moiety is in a shallow hydrophobic groove formed between TMs 8 and 10, around Leu655 (Fig. 4C). The side chains of Trp300 and Tyr461 seclude the UCB1244283 allosteric site from the UCB-J–bound site (Supplementary Fig. 6A), and binding of the modulator pushes both residues downward, reshaping the orthosteric pocket and nudging UCB-J deeper into the site by 1–2 Å, compared to its position in the UCB-J–only complex (Supplementary Fig. 6B). In addition to the movements of TMs 7, 8, and 10, which directly contact the modulator, UCB1244283 also triggers notable displacements of TMs 9, 11, and 12, compared to the UCB-J–only bound structure. Together, these helical rearrangements result in an anticlockwise rotation of the C-terminal half of SV2A (Supplementary Fig. 6C).

We mutated Phe188, Tyr461, or Ser601, key components of the allosteric site, to alanine and observed loss of allosteric potentiation by UCB1244283 (Fig. 4D). We noted that the residues forming the allosteric site in SV2A are largely conserved in SV2B, except for two residues: Leu655 and Gly659 in SV2A are substituted by Gln596 and Cys600, respectively, in SV2B (Supplementary Fig. 1C). Mutating Leu655 to alanine did not impair modulation by UCB1244283, but replacing it with glutamine to mimic the SV2B substitution abolished allosteric modulation (Fig. 4D). This observation suggests that UCB1244283 may selectively modulate SV2A but not SV2B, due to the presence of Gln596 in SV2B in the position equivalent to Leu655 in SV2A.

Earlier studies indicated that UCB1244283 can delay the dissociation of SV2A ligands such as levetiracetam and UCB30889[22]. To further investigate this effect, we assessed the kinetics of $^3$H-UCB-J dissociation from SV2A in the presence of $^1$H-UCB-J and UCB1244283. Our results show that the half-life ($t_{1/2}$) of $^3$H-UCB-J dissociation increased from $2.0 \pm 0.7$ min to $5.3 \pm 1.2$ min with UCB1244283 (Fig. 4E). These findings suggest that allosteric binding of UCB1244283 to SV2A slows ligand dissociation from the orthosteric site. Our structural data provide a mechanistic basis for this effect: UCB1244283 binds above the orthosteric site and acts as a cap

that sterically hinders ligand exit. Because the UCB1244283 site lies between the orthosteric site and the vesicular lumen, we asked whether UCB1244283 binding could hinder access of UCB-J to the orthosteric site. Indeed, when SV2A is pre-incubated with UCB1244283, the association rate of UCB-J was reduced (Fig. 4F). The effect of UCB1244283 on UCB-J association did not appear as pronounced as its effect on slowing UCB-J dissociation, which would thus result in a net increase of UCB-J occupancy at the orthosteric site (Fig. 4A). Thus, this allosteric site represents a promising pocket for designing modulators that enhance the binding of antiseizure medications and prolong their engagement with SV2A.

We also determined the structure of SV2A bound to levetiracetam and UCB1244283 at 3 Å resolution (Supplementary Figs. 2E and 4). SV2A adopts an occluded conformation, overall similar to the UCB-J/UCB1244283 structure (Supplementary Fig. 6D), with some local differences in TM7, likely due to the different interactions of levetiracetam and UCB-J at the orthosteric site. Importantly, UCB1244283 binds to the same allosteric site as observed before (Supplementary Fig. 6E), suggesting a common modulation mechanism and highlighting its capacity to influence orthosteric ligands of varying sizes and molecular features.

We attempted to capture the structure of SV2A bound to UCB1244283 alone (i.e., in the absence of orthosteric ligands) using the same concentration of UCB1244283 that yielded the two SV2A–UCB1244283 complex structures described above. However, we were unable to observe interpretable density for UCB1244283 in our cryo-EM reconstructions, and SV2A exhibited an apo conformation. This suggests that the allosteric site is not well formed in the unliganded SV2A, leading to low affinity for UCB1244283; orthosteric ligand binding induces formation of the allosteric site, resulting in increased affinity. Thus, UCB1244283 would modulate SV2A only when the orthosteric site is occupied, and such conditional engagement would confer greater specificity to the modulator, as it would be functionally inert in the absence of medications or endogenous substrates.

## Structural mechanism of padsevonil in SV2A

We further elucidated the structure of SV2A in complex with padsevonil at 2.7 Å resolution, which adopts an occluded conformation (Supplementary Fig. 2F). In our radioligand binding assay, padsevonil competitively displaced $^3$H-UCB-J from SV2A with an $IC_{50}$ of $20.9 \pm 3.1$ nM (Fig. 1), indicating strong binding affinity. Unexpectedly, the structure of SV2A with padsevonil revealed ligand density at two distinct sites (Fig. 5A and Supplementary Figs. 4 and 7A). The first padsevonil density is located in the central cavity, overlapping with the binding sites of levetiracetam and UCB-J. A second, separate density is observed at a site approximately 13 Å above the first site, in a position similar to the UCB1244283 site, as detailed below. These observations indicate that padsevonil occupies both the orthosteric and allosteric sites (Fig. 5A).

In the orthosteric site, the binding pose of padsevonil partially overlaps with that of UCB-J. The core 2-oxopyrrolidine ring is flanked by side chains of same residues as observed in the levetiracetam or UCB-J complexes (Fig. 5B). The position of the imidazothiadiazole moiety of padsevonil overlaps with that of the 3-methyl pyridine in UCB-J, but the methoxymethyl substitution in padsevonil protrudes into the interface of TMs 7 and 11, expanding its interactions (Fig. 5B) compared to UCB-J. The halogenated ethyl substitution of padsevonil is oriented toward the interface of TMs 7, 8, and 10, akin to the position of the trifluoro phenyl moiety UCB-J (Fig. 5B). Mutating residues Trp300, Trp666, or Asp670 to alanine significantly compromised padsevonil binding (Fig. 5C), confirming the critical role of these residues to SV2A sensitivity to racetam-based ligands. In addition, padsevonil binding was also sensitive to alanine substitutions of surrounding residues (Fig. 5C), consistent with our structure.

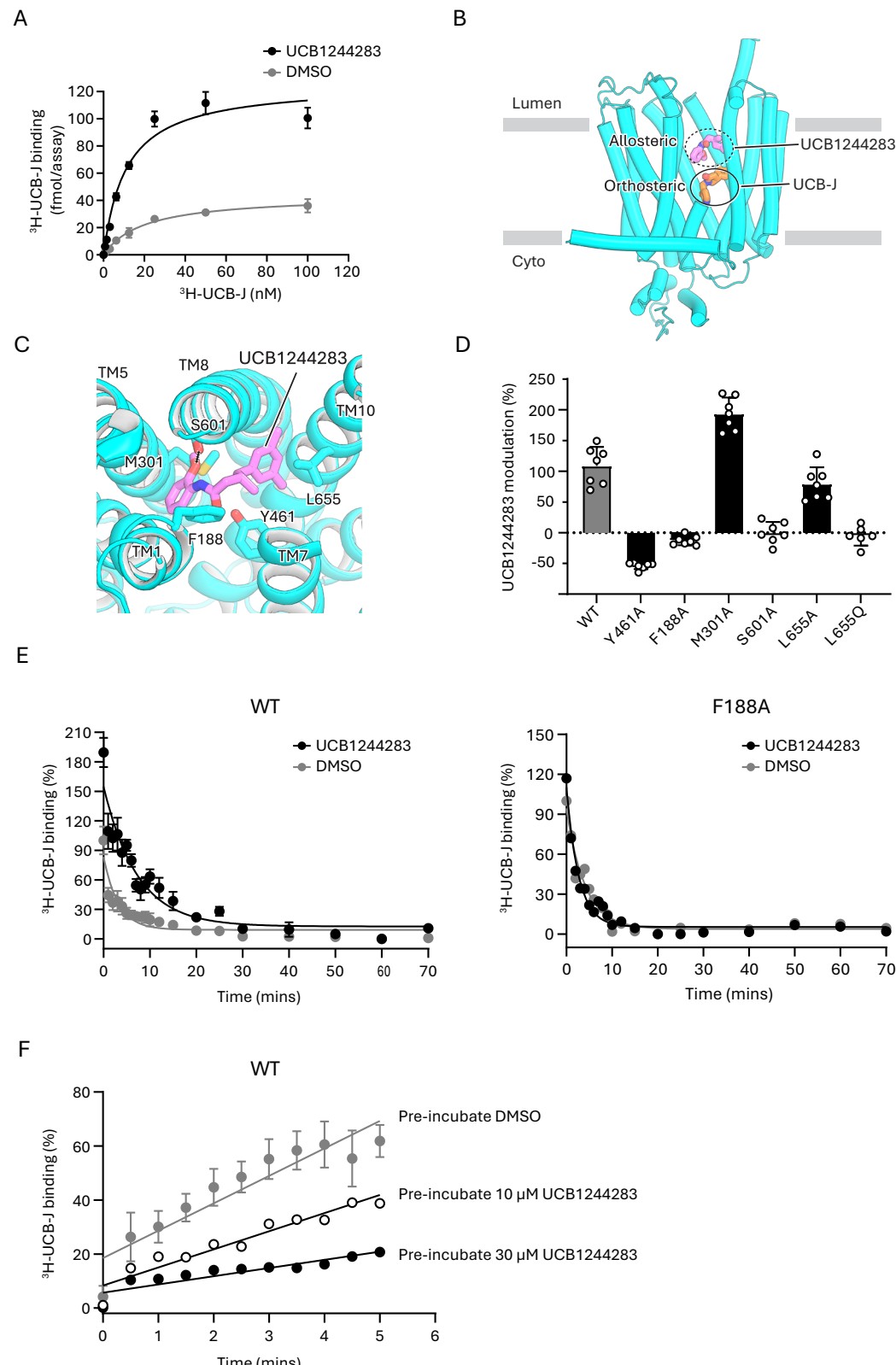

In the allosteric site, padsevonil adopts an inverted orientation relative to its binding pose in the orthosteric site (Fig. 5D and Supplementary Fig. 7A). The pocket accommodating padsevonil at this secondary site is formed by the side chains of Phe188 (TM1); Trp300 and Met301 (TM5); Tyr461 (TM7); Val600, and Ser601 (TM8); and Leu655 (TM10). Padsevonil binding is stabilized by stacking interactions between its imidazothiadiazole moiety and the side chains of

Phe188 and Leu655. Its halogenated ethyl group is oriented into a cavity lined by TM1, TM5, and TM7 (Fig. 5D). The presence of padsevonil in this site seems to cause an outward displacement of TM helices 7, 8, and 10 relative to the apo state, resulting in conformational changes to the C-domain of SV2A (Supplementary Fig. 7B).

Comparison of the padsevonil-bound SV2A structure with the UCB-J/UCB1244283 complex reveals that padsevonil partially overlaps

**Fig. 4 | Allosteric modulation of SV2A by UCB1244283. A** Steady-state binding of $^3$H-UCB-J to SV2A in the presence of 30 μM UCB1244283 (black) or 1% DMSO (gray). Data are shown as mean ± s.d.; n = 6 biological replicates. **B** Structure of SV2A (cyan) bound to UCB1244283 (magenta) in an allosteric site and UCB-J (orange) in the orthosteric site. **C** View of UCB1244283 (magenta) bound in the allosteric site of SV2A showing interactions with surrounding residues. **D** Effect of mutations on the modulatory effect of 30 μM UCB1244283 on $^3$H-UCB-J binding. Binding in the absence of UCB1244283 was defined as 0%; enhancement observed in SV2A$_{WT}$ was defined as 100%. Data are shown as mean ± s.d.; n = 6 biological replicates. **E** Dissociation kinetics of $^3$H-UCB-J from SV2A in the presence of 30 μM

UCB1244283 (black) or 1% DMSO (gray). $^3$H-UCB-J binding observed before the addition of 0.3 μM $^1$H-UCB-J in control reaction was defined as 100 %. Residual binding observed at 70 min was defined as 0%. For WT, data are shown as mean ± s.d. (n = 3 biological replicates); for F188A, data are shown as mean (n = 2) **F** Association kinetics of $^3$H-UCB-J to SV2A pre-incubated with 10 μM (white) or 30 μM (black) UCB1244283, or 1% DMSO (gray). Binding of $^3$H-UCB-J observed after 150 min in control reaction was defined as 100 %. For the DMSO group, data are shown as mean ± s.d. (n = 4 biological replicates); for the other groups, data are shown as mean (n = 2). Source data for (**A**, **D**, **E**, **F**) are provided as a Source data file.

with UCB1244283 at the allosteric site (Fig. 5E). Specifically, both the 2-pyrrolidinone moiety of padsevonil and the 2-methoxyaniline moiety of UCB1244283 occupy a pocket lined by residues from TM1 and TM5 (Fig. 5E). Consistent with this structural overlap, UCB1244283 can displace padsevonil from SV2A in a dose-dependent manner (Fig. 5F). This finding explains why UCB1244283 potentiates the binding of levetiracetam, brivaracetam[35], and UCB-J, but fails to enhance padsevonil binding[20]—likely due to direct competition at the shared allosteric region.

Beyond this overlap, padsevonil and UCB1244283 adopt divergent orientations: the imidazothiadiazole group of padsevonil is positioned between TM7 and TM10, whereas the dimethylphenyl moiety of UCB1244283 extends into a distinct pocket formed by TMs 8 and 10. This divergence suggests a notable degree of structural plasticity in the allosteric site, which may be advantageous for the design of chemically diverse modulators with tailored pharmacological properties.

## Discussion

Our structural analyses of human SV2A in the apo and multiple ligand-bound states reveal previously unobserved, ligand-induced conformational changes and identify an allosteric site in MFS proteins. Notably, we show that binding of orthosteric ligands such as levetiracetam or UCB-J triggers a conformational transition to an occluded state, characterized by inward movement of TM1 and sealing of the central cavity by Phe188. This occluded conformation may reflect a functional intermediate relevant to the transport-like behavior of SV2A, as it is analogous to occluded states commonly observed in MFS transporters. In canonical MFS transporters, alternating access is achieved through coordinated movements of transmembrane helices to gate substrate binding sites[1]. Thus, our findings support the idea that SV2A retains structural features of MFS transporters, although its endogenous substrate(s) remain to be identified. This conformational transition was not reported in prior studies of SV2A with levetiracetam[23,24], possibly due to the use of fiducial markers to stabilize the luminal domain, which may have restricted TM1 mobility. In addition, the atypical oligomerization of SV2A observed in those studies[23,25], likely resulting from detergent use, may have also affected its conformational flexibility. In contrast, our use of saposin instead of detergent micelles may have helped preserve more native-like protein dynamics. Since we obtained both apo and levetiracetam-bound states using the same SV2A construct and experimental conditions, we are able to directly compare them and infer the conformational changes that occur upon levetiracetam binding.

In addition to capturing this occluded state, our study also uncovers an allosteric site that is specific to SV2A. This site accommodates small molecules, such as UCB1244283 and padsevonil, that can modulate the binding of orthosteric ligands. In our structural analysis, UCB1244283 was bound at the allosteric site, whereas padsevonil was bound at both orthosteric and allosteric sites. We demonstrated that while UCB1244283 may partially restrict the access of orthosteric ligands, its dominant effect is to stabilize their binding by reducing the dissociation rate, enhancing occupancy at the orthosteric site. We hypothesize that padsevonil binding to the allosteric site would have a similar effect; however, because padsevonil also

binds to the orthosteric site with high affinity, it directly competes with and readily displaces orthosteric ligands, which prevents us from evaluating its modulatory effect.

UCB1244283 and padsevonil are structurally distinct compounds, yet both bind to partially overlapping regions within the SV2A allosteric site. This observation suggests an inherent versatility of that pocket, which accommodates different chemotypes by engaging both shared and ligand-specific contacts. Such flexibility could be exploited to design structurally diverse modulators, particularly by targeting the divergent extensions of the binding site to optimize selectivity and potency (Fig. 5E).

Our results are consistent with a recent study on the effect of UCB1244283 on brivaracetam[36]. Coleman and colleagues reported a structure of SV2A in complex with brivaracetam and UCB1244283, and the binding mode and mechanism of UCB1244283 they described was very similar to our findings. While their study focused on brivaracetam, our UCB-J/UCB1244283 and levetiracetam/UCB1244283 structures demonstrate that UCB1244283 may act as a versatile modulator, capable of tuning SV2A function across pharmacologically distinct compounds.

Structures of SV2B bound to padsevonil and of SV2A bound to UCB2500, a compound structurally similar to padsevonil, have been recently reported[26]. In both structures, ligand density was observed only at the orthosteric site, with no defined molecule observed at the position corresponding to the allosteric site. In the case of SV2B–padsevonil, this could be due to a substitution at the allosteric site: Leu655 in SV2A is replaced by Gln596 in SV2B, and the longer side chain of glutamine may prevent padsevonil from occupying the allosteric pocket in SV2B (Supplementary Fig. 7C). However, we were not able to detect a substantial difference in padsevonil binding in the L655Q SV2A mutant compared to wild-type SV2A protein (Supplementary Fig. 7D). We speculate that if the affinity for the allosteric site is much lower (i.e., in the micromolar range) than that for the orthosteric site (nanomolar range), the binding assay would mainly reflect binding at the orthosteric site and not effectively capture padsevonil's interaction with the allosteric site. Further studies are needed to clarify whether the Leu-to-Gln substitution in SV2B impairs padsevonil binding at the allosteric site. Padsevonil had been reported to bind all three SV2 isoforms with nanomolar affinities, but kinetics studies showed that it dissociates more slowly from SV2A compared to SV2B[20]. The isoform-specific effects of padsevonil can be explained by our findings that this compound engages both orthosteric and allosteric sites in SV2A. For the SV2A–UCB2500 structure, comparison with our SV2A–padsevonil structure reveals that the two compounds adopt similar binding poses in the orthosteric site (Supplementary Fig. 7E). However, UCB2500 has a longer butyl substitution on its pyrrolidinone moiety, which extends toward the allosteric site. This extension could cause steric clashes, which prevents simultaneous binding of two UCB2500 molecules in SV2A.

Our discovery of an allosteric site in SV2A provides mechanistic insights into ligand cooperativity and points to an underexplored regulatory layer in MFS proteins. Allosteric modulation of MFS transporters remains poorly understood at the molecular level; our findings thus offer a structural framework to inform future studies and a

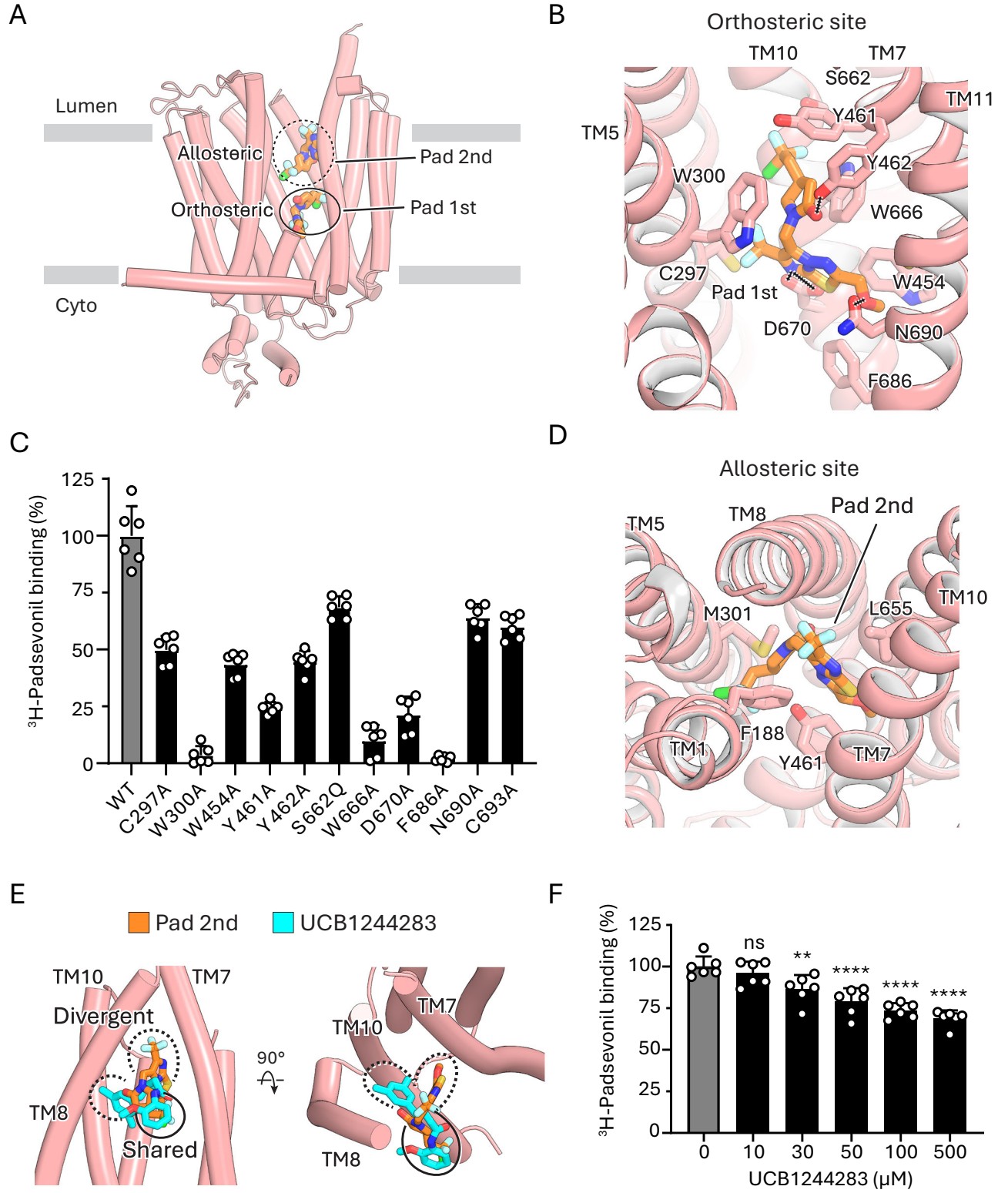

**Fig. 5 | Padsevonil binding to SV2A. A** Structure of SV2 (salmon) bound to padsevonil (orange) reveals two distinct ligand-binding sites. **B** View of padsevonil (orange) bound to the orthosteric site of SV2A, showing interactions with surrounding residues. **C** Effect of orthosteric site mutations on $^3$H-padsevonil (50 nM) binding. Binding observed with SV2A$_{WT}$ was defined as 100%. Data are shown as mean ± s.d.; n = 6 biological replicates. **D** View of padsevonil (orange) bound to the allosteric site of SV2A, showing the interactions with surrounding residues.

**E** Superposition of SV2A allosteric site bound to padsevonil or UCB1244283. **F** Dose-dependent reduction in the binding of $^3$H-padsevonil (50 nM) to SV2A by UCB1244283. Binding of $^3$H -padsevonil in the absence of UCB1244283 was defined as 100%. Data are shown as mean ± s.d.; n = 6 biological replicates. Significance values were obtained from ordinary one-way ANOVA test (n.s. not significant; **p-value = 0.0064; ****p-value < 0.0001). Source data for (**C**, **F**) are provided as a Source data file.

foundation for the rational design of SV2A-specific allosteric modulators that enhance orthosteric ligand binding, potentially paving the way for new classes of antiseizure medications.

## Methods

### Expression and purification of SV2A$_{EM}$

The SV2A$_{EM}$ construct features an N-terminal mVenus tag, a 3C protease cleavage site, and an MBP tag fused to the N-terminus of human SV2A (residues 141-742). SV2A$_{EM}$ was expressed in HEK293S GnTI− cells (ATCC, CRL-3022) using BacMam system[37]. Recombinant baculoviruses were produced by transfecting Sf9 cells (ATCC, CRL-1711) with the bacmids using TransIT Insect (Mirus). After two rounds of amplification, viruses were used for cell transduction. When HEK293S GnTI− suspension cultures were grown to a density of $4 \times 10^6$ cells/ml at 37 °C, baculoviruses (10% v/v) were added to initiate transduction. After 16–18 h, 10 mM sodium butyrate was supplemented to the cultures, and the culture temperature was reduced to 25 °C. Cells were collected at 60 h post-transduction and stored at −80 °C.

Frozen cell pellets expressing SV2A$_{EM}$ were thawed at room temperature and then resuspended in hypotonic buffer (10 mM NaCl, 1 mM MgCl$_2$, 10 mM HEPES, pH 7, protease inhibitor cocktail, and benzonase) for 20 min on ice. The cell lysate was then spun at $39,800 \times g$ for 30 min to sediment crude membranes. The membrane pellet was mechanically homogenized in 300 mM NaCl and 1 mM TCEP, 20 mM HEPES, pH 7. The suspension was solubilized in 1.2% (w/v) digitonin with 1x protease inhibitor cocktail for 90 min at 4 °C. The solubilized material was centrifuged at $39,800 \times g$ for 30 min, and the supernatant was incubated with anti-GFP nanobody resin (which also binds mVenus) for 2 h at 4 °C. The resin was washed with 10 column volumes of wash buffer A (0.05% digitonin, 150 mM NaCl, 150 mM KCl, 4 mM MgCl$_2$, 4 mM NaATP, 1 mM TCEP, and 40 mM HEPES pH 7) followed by 7 column volumes of wash buffer B (0.05% digitonin, 150 mM NaCl, 1 mM TCEP, and 20 mM HEPES pH 7). The washed resin was incubated with saposin A[38], at a molar ratio of 1:20 (SV2A$_{EM}$: saposin A). After 30 min, gamma-cyclodextrin was added to the resin to remove digitonin and allow saposin A reconstitution for overnight at 4°C. The resin was then washed with 15 column volumes of wash buffer C (150 mM NaCl, 20 mM HEPES, pH 7, and 1 mM TCEP). The washed resin was incubated with 3C protease for 2 h at a target protein to protease ratio of 40:1 (w/w) to cleave off mVenus and release the protein from the resin. The protein was eluted with wash buffer C, concentrated, and further purified by gel-filtration chromatography on a Superose 6 increase 10/300 column (Cytiva) equilibrated with buffer C. The peak fractions of protein were pooled and concentrated to ~8.0 mg/ml. While purifying the protein for UCB1244283 co-complex, 100 μM UCB1244283 was added during saposin-A reconstitution.

### Expression and purification of BONT/A

The BoNT/A pFastbac construct features a GP67 signal peptide, the receptor-binding domain of BoNT/A1 (residues 872–1296), and a C-terminal 10×His tag. BoNT/A was expressed in Sf9 cells using baculovirus. When Sf9 suspension cultures reached a density of $3 \times 10^6$ cells/ml at 27 °C, baculovirus was added at 5% (v/v). After 60 h, the medium was collected and pH-adjusted by the addition of Tris buffer (pH 8). 1 mM NiCl$_2$ and 5 mM CaCl$_2$ were added to the medium, which was then incubated with stirring for 1 h. Resulting precipitates were removed by centrifugation, and the supernatant was incubated with Ni-NTA resin. The column was washed with a high-salt buffer (500 mM NaCl, 20 mM imidazole, and 20 mM Hepes pH 7.5), followed by a wash buffer (150 mM NaCl, 20 mM imidazole, and 20 mM Hepes pH 7.5). The protein was eluted with elution buffer (150 mM NaCl, 250 mM imidazole, and 20 mM Hepes pH 7.5), concentrated, and further purified by gel-filtration chromatography on a Superdex 75 Increase 10/300 column.

### Cryo-EM sample preparation and data collection

The purified SV2A$_{EM}$ protein was incubated with either 2 mM levetiracetam (Biosynth), 100 μM UCB-J (Pharmasynth), 100 μM UCB-J with 1 mM of UCB1244283 (MuseChem), 2 mM levetiracetam with 1 mM of UCB1244283, 1 mM of UCB1244283, 200 μM padsevonil (MedChemExpress), or BoNT/A1 (molar ratio of 1:1.2) for 30 min on ice. Following this, the sample was complexed with 1.3 molar excess of MBP binding ankyrin repeat protein DARPin off7[30]. Finally, 0.4 mM fluorinated octylmaltoside was added to improve particle distribution before proceeding with cryo-EM grid preparation. Grids were prepared by applying 3.5 μl of protein sample to plasma cleaned UltrAuFoil R1.2/1.3 300 mesh grids (Quantifoil). Sample was blotted for 3.0 s at 10 °C and 100% humidity in a Vitrobot Mark IV (FEI) before plunge freezing in liquid ethane maintained at liquid nitrogen temperature. The grids were loaded onto a 300 kV Titan Krios transmission electron microscope for data collection. Raw movie stacks were recorded with a K3 camera at a physical pixel size of 0.649 Å per pixel and a nominal defocus range of 1.1–2.1 μm. The exposure time for each micrograph was 1.6–1.7 s, dose-fractionated into 60–65 frames with a dose rate of 1.01–1.08 e$^-$/pixel/s. The data collection parameters are summarized in Supplementary Table 1.

### Cryo-EM data processing

The image stacks were gain-normalized and corrected for beam-induced motion using MotionCor2[39]. Defocus parameters were estimated from motion-corrected images using cryoSPARC4[40]. Micrographs not suitable for further analysis were removed by manual inspections. Particle pickings (blob and template picker) and two-dimensional (2D) classifications were done in cryoSPARC4. Selected particles were combined with particles from blob picker and template picker, and duplicated particles were removed. Iterative three-dimensional (3D) classifications were then performed with subsequent ab initio reconstructions and heterogeneous refinements to remove suboptimal particles. Selected particles were refined using non-uniform refinement[41], followed by local refinements with soft masks covering SV2A but not MBP-DARPin to further improve map quality. The refined particles were subjected to Bayesian polishing in RELION4[42]. The polished particles were imported into cryoSPARC4, where additional refinements were performed. Mask-corrected FSC curves were calculated in cryoSPARC4, and reported resolutions are based on the 0.143 criterion. Local resolution estimations were performed in cryoSPARC4.

### Model building and refinement

An initial model of human SV2A was generated by AlphaFold[43]. This model was docked into the density maps using Chimera[44]. The model was then refined iteratively using Coot[45], ISOLDE[46], and Phenix[47]. Structural model validation was performed using Phenix and MolProbity[48]. Figures were prepared using PyMOL, Chimera, and ChimeraX.

### Membrane preparation for radioligand binding experiments

Ligand binding studies were carried out using crude membranes prepared from HEK293 cells (ATCC, CRL-1573) overexpressing mVenus-SV2A$_{WT}$ or SV2A$_{EM}$. Membranes were prepared either from transiently transfected cells or from baculovirus-infected cells as described previously[28]. Cells were washed with DPBS (Ca$^{2+}$ and Mg$^{2+}$ free), followed by resuspending in cold SH buffer (10 mM HEPES-Tris pH 7.4, 320 mM sucrose) containing protease inhibitor cocktail (MedChemExpress) to a density of $10^7$ cells/ml. This cell suspension was homogenized by passing through a bead homogenizer (Isobiotec) with 10-micron clearance bead on ice. The lysate was then centrifuged at $4000 \times g$ for 5 min at 4 °C, and the supernatant was aliquoted, frozen in liquid nitrogen, and stored at −80 °C. The mVenus fluorescence estimated using fluorescence size-exclusion chromatography (FSEC) was

used to normalize expression levels of mVenus-SV2A mutants studied as detailed previously[28]. For all binding studies, membranes containing 5–30 mg protein (estimated by BCA method) per reaction were used.

### Dose-response studies

To obtain dose-response curves of studied SV2A ligands, membranes diluted in 0.2 ml binding buffer (50 mM Tris-Cl, pH 7.4, and 2 mM $MgCl_2$) were incubated with indicated concentrations of ligands for 30 min at 4 °C. To these, 10 nM $^3$H-UCB-J (67.2 Ci/mmol, RC Tritec AG) was added and incubated for another 90 min at 4 °C. The binding reaction was arrested by trapping membranes on Unifilter-GF/C 96-well filter plates (Revvity) pre-soaked with 0.1%(w/v) poly-ethyleneimine (PEI). Membranes were then washed four times with 0.2 ml of cold binding buffer, dried, and 40 μl of scintillation liquid was added to each well. Radioactivity was monitored in MicroBeta2 plate counter (Revvity). Estimates for $IC_{50}$ were obtained from fitting the data to three-parameter dose-response curves using GraphPad Prism 10.

### Saturation binding studies

Saturation binding studies were performed by incubating SV2A expressing membranes with indicated concentrations of $^3$H-UCB-J or $^3$H-padsevonil (1.2 Ci/mmol, Moraveck Inc) for 2 h in 0.2 ml of binding buffer at 4 °C. In assays involving UCB1244283, it was added to a final concentration of 30 μM from its 3 mM stock (DMSO) after 30 min incubation with the radioligand. An equal volume of dimethyl sulfoxide (DMSO) was added to control reactions. Reactions were arrested after 2 h by trapping the membranes on a Unifilter-GF/C 96-well filter plate pre-soaked with 0.1%(w/v) polyethyleneimine. Membranes were washed four times with 0.2 ml of cold binding buffer, dried, and 40 μl of scintillation liquid was added to each well. Radioactivity was monitored in MicroBeta2 plate counter. Residual binding observed in presence of either 10 μM $^1$H-UCB-J or 10 μM $^1$H-padsevonil was considered as background and subtracted from total binding. Estimates for $K_d$ and $B_{max}$ were obtained by fitting the data to one site-specific binding curve using GraphPad Prism 10.

### Dissociation kinetics studies

Radioligand dissociation kinetics was studied by incubating $SV2A_{EM}$ WT or $SV2A_{EM}$ F188A expressing membranes with 10 nM $^3$H-UCB-J for 2 h at 4 °C. UCB1244283 was added to a final concentration of 30 μM after 30 min incubation with the radioligand, and an equal volume of DMSO was added to the control reaction. After 2 h of incubation, ligand dissociation was initiated with 0.3 μM $^1$H-UCB-J and radioactivity was monitored at indicated time points by trapping equal volume of membranes on PEI-treated Unifilter-GF/C 96-well filter plates. Dissociation half-life ($t_{1/2}$) estimates were obtained by fitting the data to one-phase exponential decay using GraphPad Prism 10.

### Association kinetics studies

Radioligand association kinetics assays to determine the effect of UCB1244283 on the association rate of UCB-J binding were carried out using $SV2A_{EM}$ membranes. Membranes diluted in binding buffer were incubated on ice with either 10 or 30 μM of UCB1244283 or DMSO (control) for 30 min. Following this, $^3$H-UCB-J was added to a final concentration of 5 nM and binding was monitored at indicated time points by trapping the membranes on PEI-treated Unifilter-GF/C 96-well plates and washing off excess radioligand with cold binding buffer. Residual radioligand binding observed in presence of 10 μM of $^1$H-UCB-J was considered as background binding.

### Reporting summary

Further information on research design is available in the Nature Portfolio Reporting Summary linked to this article.

## Data availability

The cryo-EM maps have been deposited in the Electron Microscopy Data Bank (EMDB) under accession codes EMD-70562 (apo); EMD-70563 (levetiracetam); EMD-70564 (UCB-J); EMD-70565 (UCB-J + UCB1244283); EMD-71812 [https://www.ebi.ac.uk/pdbe/entry/emdb/EMD-71812] (levetiracetam + UCB1244283); and EMD-70566 [https://www.ebi.ac.uk/pdbe/entry/emdb/EMD-70566] (padsevonil). The atomic coordinates have been deposited in the Protein Data Bank (PDB) under accession codes 9OKF (apo); 9OKG (levetiracetam); 9OKH (UCB-J); 9OKI (UCB-J + UCB1244283); 9PRS (levetiracetam + UCB1244283); and 9OKJ (padsevonil). Source data are provided with this paper.

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

## Acknowledgements

The authors thank scientists in the Cryo-EM Center of St. Jude Children's Research Hospital for their support in data collection. The authors thank members of the Lee lab for helpful discussions; Dr. I. Chen for editing the manuscript. The authors thank the Roussel lab at St. Jude for sharing space for radioisotope experiments. This work is supported by the National Institutes of Health (R01NS133147) and ALSAC.

## Author contributions

S.P. performed all the experiments. X.C. and L.N.N. assisted with functional assays. Y.D., Y.N., C.G., and F.L. contributed to structural analysis. C.-H.L. conceived the research and supervised the project. S.P. and C.-H.L. wrote the manuscript with input from all authors.

## Competing interests

The authors declare no competing interests.
