## [Transparent Peer Review file · Nature Communications]

Structural pharmacology of SV2A reveals an allosteric modulation mechanism in the major facilitator superfamily

Corresponding Author: Dr Chia-Hsueh Lee

Version 0:

Reviewer comments:

Reviewer #1

(Remarks to the Author)

This paper complements very well a recent study by Coleman and co-workers that was published as a preprint in bioRxiv (<https://doi.org/10.1101/2025.05.05.652227>). Using brivaracetam (BRV), they also structurally and functionally characterized SV2A binding of UCB1244283, which has been described as a positive allosteric modulator of antiseizure medications. In the present manuscript UCB-J is used instead of BRV and very similar results are obtained. Furthermore, they observe an outward-facing occluded receptor conformation for SV2A bound to levetiracetam (LEV). This is in contrast to other SV2A-LEV complex structures that reported an outward-facing open conformation. The authors also elucidated the first apo-state structure of SV2A and the first complex structure of SV2A bound to UCB-J. Most notably, the authors observe binding of padosenovil (PSL) to both the allosteric and the primary binding site of SV2A.

I feel that the current findings are too preliminary to recommend publication in Nature Communications. However, the manuscript should be seriously considered for publication if the authors manage to provide additional experimental evidence and discuss their findings in the context of the above-mentioned preprint.

Major points:

- The findings of the present study need to be discussed in the context of the above-mentioned preprint, which is currently not cited.
 - The authors don't see the luminal domain in their SV2A structures. This is in contrast to the findings of five other groups that published cryo-EM structures of SV2 proteins. Some of these structures were published in complex with the same compounds (LEV, PSL) described in the present study. This is worrisome. To confirm the structural integrity of the fusion protein, its structure should be determined in the presence of the receptor-binding domain of a clostridial toxin. This seems particularly important because the protein contains MBP, which can act as a chaperone.
 - Based on the data presented in this paper and the results published in the literature, UCB1244283 binds to SV2A without BRV or LEV. UCB1244283 alone has a clear protective effect against tonic and clonic convulsions (Daniels V, Wood M, Leclercq K, Kaminski RM, Gillard M. Modulation of the conformational state of the SV2A protein by an allosteric mechanism as evidenced by ligand binding assays. *Br J Pharmacol.* 2013 Jul;169(5):1091-101. doi: 10.1111/bph.12192. PMID: 23530581; PMCID: PMC3696331.). For a more complete understanding of how UCB1244283 modulates the binding of antiseizure drugs to SV2A, the cryo-EM structure of an SV2A-UCB1244283 complex should be determined. Is UCB1244283 only binding to the allosteric site or also the primary binding site? Are there conformational changes without BRV or UCB-J present?
 - Like reported for BRV in the above-mentioned preprint, UCB1244283 prevents dissociation of UCB-J by blocking the luminal gate. Is UCB1244283 blocking or delaying the binding of UCB-J if added first? I think such experiments are important to obtain further insights into the action of UCB1244283.
 - To my knowledge, UCB-J is not an approved antiseizure medication. Why did the authors not use BRV to study the modulation by UCB1244283?
 - PSL binds to both the allosteric and the primary binding site of SV2A. This raises the question whether the allosteric site is not rather just a second binding site. This would be consistent with the observation that UCB1244283 alone has a protective effect on seizures.
 - Does PSL modulate the binding of LEV or BRV to SV2A? Based on the findings of the present study, such experiments need to be included.
- Minor points:
- SV2s belong to the SLC22 family. Is the allosteric binding site conserved in some of these proteins like for example the OCTs?

- Because no information is provided, I guess that the allosteric binding site of SV2A is not conserved in all MFS transporters. Therefore, the title is misleading and should be changed.
- In contrast to other complex structures, LEV binds SV2A in an occluded conformation of the luminal half. Do the authors also see the open form? How does the occluded conformation compare to the SV2 structures bound to UCB-2500 or PSL?

Reviewer #2

(Remarks to the Author)

The paper reports the structures of SV2A in apo and in complexes with levetiracetam, UCB-J, UCB1244283 and padsevonil. Some of these compounds are FDA-approved anti-epilepsy drugs. The cryo-EM structures have revealed the binding sites of the compounds, and the binding pocket for each ligand is validated using mutagenesis and binding assays. In particular, an allosteric site is identified for UCB1244283, a compound that is previously known for being a potentiator for ligand binding at the main binding site in the transporter's central cavity. As few allosteric modulators have been identified for MFS proteins, this finding is quite significant. Overall, these experiments have been carefully designed and carried out, and the manuscript clearly written.

One problem with the work is that the physiological role of SV2A in the cell is unclear. Located in the synaptic vesicles, the protein is abundant. SV2A is presumably an MFS transporter, but neither its endogenous substrate nor the mechanism of transport is known. How the drugs studied in this work exert their pharmacological activity is also unknown. Are the drugs inhibitors, activators or folding correctors? Therefore, the allosteric modulation described here does not have a cellular context. Still, the work reported in this paper represents significant progress of the field and should therefore be published despite the weakness.

Line 107, "indicating that the construct modification does not impair function(Fig. 1A)." The data in Fig. 1A only indicated that the truncation-fusion modification did affect drug binding at the central cavity. It provided no evidence that the modification did not affect the function of the transporter.

The Coulomb potential densities for each of the bound ligands and the surrounding residues should be shown clearly. The way they are displayed as in Fig. S3 is inadequate.

Fig. S1C. The amino acid sequence letters are too small to read. The three parts of the figure should be rearranged to make the sequences in Panel C readable.

In the SV2A-Lev structure (Figs. 2 & S4), the N-terminal half of TM1 is bent, a significant difference from a previously published structure. As the N-terminus is where the construct is truncated and fused with a MBP domain, how can we rule out the possibility that the helix change is partially caused by the construction modification?

Lines 144-145, "UCB-J binds to SV2AEM with strong affinity and exhibits an IC50 of 4.1 ± 0.4 nM." What specific activity of the transporter does the compound inhibit?

The transporter conformations to which each ligand binds to should be explicitly stated.

Reviewer #3

(Remarks to the Author)

Pidathala et al. present five cryo-EM structures of synaptic vesicle glycoprotein 2A (SV2A) in apo states and bound to levetiracetam, UCB-J alone, UCB-J/UCB1244283, and padsevonil – all compounds of varying therapeutic use and interest. To do so, they applied a previously reported protein engineering strategy to generate a SV2A construct facilitating structural studies, which they demonstrate does not significantly alter its pharmacological binding properties. All four inhibitors bind to and presumably stabilize conformationally distinct states relative to one another and the lumen-facing apo state: levetiracetam-bound and UCB-J-bound SV2A is occluded and partially occluded, respectively, while UCB-J/UCB1244283-bound and padsevonil-bound SV2A adopt altered lumen-facing states with occupancy of a newly identified allosteric site.

Comparative analyses between their five structures and previously published work on SV proteins, coupled with radiolabeled ligand-binding assays and mutagenesis, provide insight towards the underlying mechanisms of SV2A pharmacological modulation. The identification of the allosteric site sheds light on the modulation of SV2A, and the discovery that the same ligand can bind to both orthosteric and allosteric sites is novel and highly interesting.

Overall, the authors present a well-executed study of the structural pharmacology of SV2A and offer important new insights into the allostery of MFS-fold proteins. This work makes important contribution to the field.

There are several points that need to be addressed.

Main Points:

1) Some of the reported mutants seem to unexpectedly alter inhibitor sensitivities, e.g., F686A and N690A to levetiracetam. F686A, for instance, is depicted as reasonably distant from the orthosteric site – yet consistently abolishes binding across

inhibitors. N690A is posed to eliminate potential side-chain interactions between levetiracetam, for instance, yet further enhances its binding. Discussion of these observations would be appreciated.

2) Broadly, further discussion and rationalization of the differential response of key point mutants across inhibitors/conformations would be beneficial. For instance, W454A and Y461A seem largely neutral to UCB-J binding but abolish levetiracetam sensitivity – which again seems unexpected based on the inhibitors' structure and modeled pose. Could this be related to differences assumed in an occluded versus partially occluded state? Regardless, the authors should further comment on this.

3) The authors mention differences in conservation between allosteric sites in SV2A and SV2B as a possible explanation for UCB1244283's selective modulation of the A isoform, but is the compound known to be a selective modulator to begin with? Please clarify.

4) It seems to be difficult to tease out the conformational differences caused by allosteric modulator and the orthosteric ligand since both binds to the transporter. Please clarify.

5) As proposed in the discussion, can the authors validate the importance of the side-chain substitutions in 2A vs. 2B with respect to padsevonil allosteric binding?

6) The finding that padsevonil binds to both sites are unexpected and novel. Is there any indication to support that two molecules bind to one transporter (for example, Hill coefficient of the binding)?

Minor Points:

- Fig. 2A/3A/5B: label of residue 690 is incorrect; N, not Q
- Line 57: "MFS member remain limited" – should be 'members'
- Line 196: "forming allosteric site" – missing article
- Line 490: Fig. 1C legend – the modeled ICHs are perpendicular to the TMs, not parallel
- Line 527: "n =6 biological replicates." – missing space
- Line 530: missing punctuation
- Line 554: Fig. S2 legend – formatting and missing spaces
- Fig. S3 – can the authors clearly label which models correspond to each map?
- Fig. S5A/S6A – style of cartoon used for atom models are inconsistent with the rest of the manuscript and, visually, hard to distinguish

Reviewer #4

(Remarks to the Author)

Version 1:

Reviewer comments:

Reviewer #1

(Remarks to the Author)

With one exception, the authors have addressed all of my comments satisfactorily. However, I still find the title confusing and suggest changing it to something like "Structural pharmacology of SV2A reveals an allosteric modulation mechanism" or "Cryo-EM structures of SV2A reveal an allosteric modulation mechanism".

Reviewer #2

(Remarks to the Author)

The authors have addressed all my concerns adequately. I now support the publication of the manuscript.

Reviewer #3

(Remarks to the Author)

Thank you for addressing the previous comments. No further questions. The manuscript is ready for publication.

Reviewer #4

(Remarks to the Author)

POINT-BY-POINT RESPONSE

Reviewer #1 (Remarks to the Author):

This paper complements very well a recent study by Coleman and co-workers that was published as a preprint in bioRxiv (<https://doi.org/10.1101/2025.05.05.652227>). Using brivaracetam (BRV), they also structurally and functionally characterized SV2A binding of UCB1244283, which has been described as a positive allosteric modulator of antiseizure medications. In the present manuscript UCB-J is used instead of BRV and very similar results are obtained. Furthermore, they observe an outward-facing occluded receptor conformation for SV2A bound to levetiracetam (LEV). This is in contrast to other SV2A-LEV complex structures that reported an outward-facing open conformation. The authors also elucidated the first apo-state structure of SV2A and the first complex structure of SV2A bound to UCB-J. Most notably, the authors observe binding of padosenovil (PSL) to both the allosteric and the primary binding site of SV2A.

I feel that the current findings are too preliminary to recommend publication in Nature Communications. However, the manuscript should be seriously considered for publication if the authors manage to provide additional experimental evidence and discuss their findings in the context of the above-mentioned preprint.

We thank the reviewer for the constructive comments. We have performed all the requested experiments, which have strengthened the manuscript.

Major points:

- The findings of the present study need to be discussed in the context of the above-mentioned preprint, which is currently not cited.

We have cited and discussed the preprint by Coleman and co-workers in the revision. The work became available around the time we submitted our manuscript, and we are pleased to see how the two studies complement each other, as they independently uncover the structural basis of SV2A modulation mechanisms.

- The authors don't see the luminal domain in their SV2A structures. This is in contrast to the findings of five other groups that published cryo-EM structures of SV2 proteins. Some of these structures were published in complex with the same compounds (LEV, PSL) described in the present study. This is worrisome. To confirm the structural integrity of the fusion protein, its structure should be determined in the presence of the receptor-binding domain of a clostridial toxin. This seems particularly important because the protein contains MBP, which can act as a chaperone.

We regret that our wording was unclear. We do observe the luminal domain, but due to its flexibility, this region is poorly resolved in the final reconstruction. We have revised the relevant passages to improve clarity. In the new Fig. S3A, we show the density of the luminal domain in the 2D classification and in the unsharpened map, and the density we observe is comparable to, if not better than, that reported by Qu and co-workers (PMID: 38773074, who, like us, did not use fiducial markers).

Nevertheless, we have performed the experiment suggested by the reviewer and determined the structure of SV2A in complex with the receptor-binding domain of BoNT/A toxin at 3.4 Å resolution (Fig. S3B). Our structure is essentially identical to the SV2A-toxin cryo-EM structure reported by Yamagata and co-workers (PMID: 38637505) (shown in Fig. S3C and below), providing strong evidence that our construct modification did not compromise the structural integrity of SV2A. We have added these data and discussion into the revised manuscript.

- Based on the data presented in this paper and the results published in the literature, UCB1244283 binds to SV2A without BRV or LEV. UCB1244283 alone has a clear protective effect against tonic and clonic convulsions (Daniels V, Wood M, Leclercq K, Kaminski RM, Gillard M. Modulation of the conformational state of the SV2A protein by an allosteric mechanism as evidenced by ligand binding assays. *Br J Pharmacol.* 2013 Jul;169(5):1091-101. doi: 10.1111/bph.12192. PMID: 23530581; PMCID: PMC3696331.). For a more complete understanding of how UCB1244283 modulates the binding of antiseizure drugs to SV2A, the cryo-EM structure of an SV2A-UCB1244283 complex should be determined. Is UCB1244283 only binding to the allosteric site or also the primary binding site? Are there conformational changes without BRV or UCB-J present?

We would like to clarify that our data suggest that UCB1244283 alone may bind to SV2A, but with low affinity. In our original submission, we mentioned our unsuccessful attempt to determine the structure of the SV2A-UCB1244283 complex, using the same concentration of UCB1244283 as that used to obtain the UCB-J/UCB1244283 and LEV/UCB1244283 complexes, and we could not increase it further due to poor solubility. These observations are consistent with the allosteric site not being well formed (i.e., very low affinity) in unliganded SV2A; orthosteric ligand binding induces the formation of the allosteric site (increased affinity). We have expanded the text to make these points clearer.

The in vivo protective effects of UCB1244283 could arise from its modulation of SV2A when an unidentified endogenous substrate is bound to the transporter, or from other off-target effects.

- Like reported for BRV in the above-mentioned preprint, UCB1244283 prevents dissociation of UCB-J by blocking the luminal gate. Is UCB1244283 blocking or delaying the binding of UCB-J if added first? I think such experiments are important to obtain further insights into the action of UCB1244283.

Thank you for the suggestion, and we have performed the recommended experiment (Fig. 4F of the revision and below). When SV2A is pre-incubated with UCB1244283, the association rate of UCB-J was reduced. This observation is consistent with our structural data, which shows that the UCB1244283 binds to a site between the orthosteric site and the vesicular lumen. The binding of UCB1244283 would therefore hinder UCB-J from accessing the orthosteric site, and we have included additional discussion in the revision.

- To my knowledge, UCB-J is not an approved antiseizure medication. Why did the authors not use BRV to study the modulation by UCB1244283?

We could not use BRV (Brivaracetam), since it is classified as a controlled substance by the U.S. Drug Enforcement Administration and we lack the necessary license.

To address this comment from the reviewer, we determined a new structure of SV2A in complex with LEV (an approved medication) and UCB1244283 at 3 Å resolution (Figs. S2D and S4). The data show that UCB1244283 binds to the same site (Fig. S6D, E; part of it shown in Fig. below) whether it is paired with UCB-J, LEV (reported by us) or BRV (reported by Coleman and co-workers), suggesting a common modulation mechanism and highlighting its capacity to influence orthosteric ligands of varying sizes and molecular features.

- PSL binds to both the allosteric and the primary binding site of SV2A. This raises the question whether the allosteric site is not rather just a second binding site. This would be consistent with the observation that UCB1244283 alone has a protective effect on seizures.

Thank you for the comment. We observed that PSL binds to both the allosteric and primary sites simultaneously, rather than independently. Given the positioning of the bound PSL, we lean toward a model in which the second PSL allosterically modulates PSL binding at the primary site, similar to how UCB1244283 modulates the binding of UCB-J. Regarding UCB1244283, results from our group and others clearly indicate that it can allosterically modulate drug binding, rather than simply binding to a different site. The in vivo protective effect of UCB1244283 alone could arise from its modulation of SV2A when an unidentified endogenous substrate is bound to the transporter, or from other off-target effects.

- Does PSL modulate the binding of LEV or BRV to SV2A? Based on the findings of the present study, such experiments need to be included.

We appreciate this comment. It is not feasible to evaluate the modulatory effect of PSL on the binding of LEV or BRV, because PSL also binds to the orthosteric site, with much higher affinity than LEV (>1000-fold) or BRV (~100-fold), and directly competes with and readily displaces them (see Fig. below, for LEV, and PMID: 31619465 for BRV). This competitive effect of PSL on LEV and BRV has also been described in the literature (PMID: 31619465). We have discussed this limitation in the revised manuscript.

Minor points:

- SV2s belong to the SLC22 family. Is the allosteric binding site conserved in some of these proteins like for example the OCTs?

Among the four key residues in the allosteric site (F188, Y461, S601, and L655), F188 and Y461 are conserved in OCT1 and OCT2; S601 is not conserved, and L655 is replaced by an alanine in OCT1 and OCT2 (Fig. below). Notably, we have shown that the SV2A L655A mutant can still be modulated by UCB1244283 (Fig. 4D).

■ SV2A apo (this study) ■ OCT1 (PDB: 8ET6)
■ OCT2 (PDB: 8ET9)

- Because no information is provided, I guess that the allosteric binding site of SV2A is not conserved in all MFS transporters. Therefore, the title is misleading and should be changed.

As described above, some of the key residues at this site are conserved in OCTs. However, it is unlikely that any single site, orthosteric or allosteric, is conserved across the entire MFS transporter family, and we fully

expect that other allosteric sites/mechanisms exist in other MFS members. Our title is meant to reflect that we are studying *a* mechanism of allosteric modulation, not the universal mechanism across the family.

- In contrast to other complex structures, LEV binds SV2A in an occluded conformation of the luminal half. Do the authors also see the open form? How does the occluded conformation compare to the SV2 structures bound to UCB-2500 or PSL?

Thank you for the questions. We do not observe an open conformation in our LEV dataset. The LEV-bound and UCB-2500 structures are similar. While there are subtle movements in TM1 and Phe188, the transporter remains occluded in both cases (Fig. A below). When comparing the LEV and PSL structures, we observe additional movements in TM7–TM11 in the latter (Fig. B below), likely due to the binding of the second PSL molecule.

Reviewer #2 (Remarks to the Author):

The paper reports the structures of SV2A in apo and in complexes with levetiracetam, UCB-J, UCB1244283 and padsevonil. Some of these compounds are FDA-approved anti-epilepsy drugs. The cryo-EM structures have revealed the binding sites of the compounds, and the binding pocket for each ligand is validated using mutagenesis and binding assays. In particular, an allosteric site is identified for UCB1244283, a compound that is previously known for being a potentiator for ligand binding at the main binding site in the transporter's central cavity. As few allosteric modulators have been identified for MFS proteins, this finding is quite significant. Overall, these experiments have been carefully designed and carried out, and the manuscript clearly written.

One problem with the work is that the physiological role of SV2A in the cell is unclear. Located in the synaptic vesicles, the protein is abundant. SV2A is presumably an MFS transporter, but neither its endogenous substrate nor the mechanism of transport is known. How the drugs studied in this work exert their pharmacological activity is also unknown. Are the drugs inhibitors, activators or folding correctors? Therefore, the allosteric modulation described here does not have a cellular context. Still, the work reported in this paper represents significant progress of the field and should therefore be published despite the weakness.

Thank you for your positive comments and thoughtful feedback, which helped us improve our manuscript.

Line 107, "indicating that the construct modification does not impair function (Fig. 1A)." The data in Fig. 1A only indicated that the truncation-fusion modification did affect drug binding at the central cavity. It provided no evidence that the modification did not affect the function of the transporter.

The reviewer is correct. To avoid confusion, we have revised the sentence to "indicating that the construct modification does not impair drug binding."

The Coulomb potential densities for each of the bound ligands and the surrounding residues should be shown clearly. The way they are displayed as in Fig. S3 is inadequate.

Thank you for the suggestion. In the original Fig. S3, the densities for each bound ligand and the surrounding residues were in different colors, causing confusion. We have now redrawn Fig. S3 (now Fig. S4) and clearly labeled which model corresponds to each dataset.

Fig. S1C. The amino acid sequence letters are too small to read. The three parts of the figure should be rearranged to make the sequences in Panel C readable.

We regret the inconvenience and have redrawn this panel to improve readability.

In the SV2A-Lev structure (Figs. 2 & S4), the N-terminal half of TM1 is bent, a significant difference from a previously published structure. As the N-terminus is where the construct is truncated and fused with a MBP domain, how can we rule out the possibility that the helix change is partially caused by the construction modification?

We appreciate the question and would like to clarify that it is actually the C-terminal half of TM1 (facing the lumen) that has a different conformation in our structure; the N-terminal half of TM1 (facing the cytosol) adopts a structure similar to that reported by other groups. Furthermore, our truncation occurs before ICH0, the first intracellular helix, and is located far from TM1. Thus, the bent region in the C-terminal half of TM1 is

spatially distant from the truncation site, making it unlikely that the observed helical change is caused by the construct modification.

Lines 144-145, "UCB-J binds to SV2AEM with strong affinity and exhibits an IC₅₀ of 4.1 ± 0.4 nM." What specific activity of the transporter does the compound inhibit?

Thank you for your question. In this experiment, we used unlabeled (cold) UCB-J to compete with ³H-UCB-J. To improve clarity, we have revised the sentence to explicitly indicate that this was a radioligand displacement assay.

The transporter conformations to which each ligand binds to should be explicitly stated.

We appreciate the suggestion and have explicitly stated the conformations of all structures.

Reviewer #3 (Remarks to the Author):

Pidathala et al. present five cryo-EM structures of synaptic vesicle glycoprotein 2A (SV2A) in apo states and bound to levetiracetam, UCB-J alone, UCB-J/UCB1244283, and padsevonil – all compounds of varying therapeutic use and interest. To do so, they applied a previously reported protein engineering strategy to generate a SV2A construct facilitating structural studies, which they demonstrate does not significantly alter its pharmacological binding properties. All four inhibitors bind to and presumably stabilize conformationally distinct states relative to one another and the lumen-facing apo state: levetiracetam-bound and UCB-J-bound SV2A is occluded and partially occluded, respectively, while UCB-J/UCB1244283-bound and padsevonil-bound SV2A adopt altered lumen-facing states with occupancy of a newly identified allosteric site.

Comparative analyses between their five structures and previously published work on SV proteins, coupled with radiolabeled ligand-binding assays and mutagenesis, provide insight towards the underlying mechanisms of SV2A pharmacological modulation. The identification of the allosteric site sheds light on the modulation of SV2A, and the discovery that the same ligand can bind to both orthosteric and allosteric sites is novel and highly interesting.

Overall, the authors present a well-executed study of the structural pharmacology of SV2A and offer important new insights into the allostery of MFS-fold proteins. This work makes important contribution to the field.

We sincerely thank the reviewer for their thoughtful feedback, which helped us clarify and improve our interpretation of the data.

There are several points that need to be addressed.

Main Points:

1) Some of the reported mutants seem to unexpectedly alter inhibitor sensitivities, e.g., F686A and N690A to levetiracetam. F686A, for instance, is depicted as reasonably distant from the orthosteric site – yet consistently abolishes binding across inhibitors. N690A is posed to eliminate potential side-chain interactions between levetiracetam, for instance, yet further enhances its binding. Discussion of these observations would be appreciated.

Both F686 and N690 have their side chains oriented toward the orthosteric site, which is why we chose to mutate them, but they are indeed somewhat distant from the ligand, as the reviewer points out. For example, the distance between N690 and levetiracetam is 4.4 Å (Fig. below), suggesting that it may not directly interact with the ligand. These two residues may be involved in maintaining the integrity of the ligand-binding pocket or in transporter gating, and thus their mutations could indirectly influence ligand binding. We have added this discussion to the revised manuscript.

2) Broadly, further discussion and rationalization of the differential response of key point mutants across inhibitors/conformations would be beneficial. For instance, W454A and Y461A seem largely neutral to UCB-J binding but abolish levetiracetam sensitivity – which again seems unexpected based on the inhibitors' structure and modeled pose. Could this be related to differences assumed in an occluded versus partially occluded state? Regardless, the authors should further comment on this.

We appreciate the suggestion and have expanded discussion of the mutants in the revision. Levetiracetam is a small molecule, and thus a single mutation to a contacting residue could have a profound effect on its binding. In contrast, UCB-J is bulkier and interacts with a larger surface area of the transporter; in the W454A or Y461A mutants, additional contacts may help maintain strong binding for UCB-J. It is also possible that certain mutations have secondary effects that influence the transition between different conformational states of the transporter and differentially affect the binding of UCB-J and levetiracetam.

3) The authors mention differences in conservation between allosteric sites in SV2A and SV2B as a possible explanation for UCB1244283's selective modulation of the A isoform, but is the compound known to be a selective modulator to begin with? Please clarify.

Thank you for this question. Before our study, there were no reports on whether UCB1244283 could act on other isoforms, and our mutagenesis results suggest that it may not work on SV2B. We have rephrased the passage to avoid confusion, and it now reads

“This observation suggests that UCB1244283 may selectively modulate SV2A but not SV2B, due to the presence of Gln596 in SV2B in the position equivalent to Leu655 in SV2A.”

4) It seems to be difficulty to tease out the conformational differences caused by allosteric modulator and the orthosteric ligand since both binds to the transporter. Please clarify.

For padsevonil, which binds to both allosteric and orthosteric sites, the reviewer is correct that we are currently unable to distinguish whether the observed conformational changes result from binding at either site or both sites. We were therefore cautious with our phrasing to avoid overinterpretation. On the other hand, in the case of UCB-J and levetiracetam, our four structures (UCB-J alone, levetiracetam alone, UCB-J with UCB1244283, and levetiracetam with UCB1244283) allow us to distinguish conformational changes due to the binding of the allosteric modulator (Fig. S6C).

5) As proposed in the discussion, can the authors validate the importance of the side-chain substitutions in 2A vs. 2B with respect to padsevonil allosteric binding?

For the revision, we have performed ³H-padsevonil binding experiments for WT and L655Q SV2A and did not detect a substantial difference (Fig. S7D, also shown in Fig. below). We speculate that if the affinity for the allosteric site is much lower (i.e., in the micromolar range) than that for the orthosteric site (nanomolar range), the binding assay would mainly reflect binding at the orthosteric site and not effectively capture padsevonil's interaction with the allosteric site. We were unable to use higher concentrations of padsevonil due to the limited stock concentration of ³H-padsevonil. We have added this discussion to the revised manuscript, stating this limitation and noting that future studies are needed to clarify whether the Leu-to-Gln substitution in SV2B affects padsevonil binding.

6) The finding that padsevonil binds to both sites are unexpected and novel. Is there any indication to support that two molecules bind to one transporter (for example, Hill coefficient of the binding)?

We were able to fit the padsevonil binding data to a one-site binding curve (shown in Fig. above), which is consistent with previous literature showing that the Hill coefficient of padsevonil is close to 1 (PMID: 31619465). We believe this may be because the binding assay primarily measures binding of padsevonil to the orthosteric site (as discussed above). We note that it is not uncommon for binding assays to miss or fail to resolve multiple binding sites in transporters. For example, binding experiments with citalopram and serotonin transporter yielded a one-site binding curve, whereas structural studies have shown that citalopram binds to both the allosteric and orthosteric sites (PMID: 27049939)

Minor Points:

- Fig. 2A/3A/5B: label of residue 690 is incorrect; N, not Q
- Line 57: “MFS member remain limited” – should be ‘members’
- Line 196: “forming allosteric site” – missing article
- Line 490: Fig. 1C legend – the modeled ICHs are perpendicular to the TMs, not parallel
- Line 527: “n =6 biological replicates.” – missing space
- Line 530: missing punctuation
- Line 554: Fig. S2 legend – formatting and missing spaces

We thank the reviewer and have corrected all the above points.

- Fig. S3 – can the authors clearly label which models correspond to each map?

We have redrawn Fig. S3 (now Fig. S4) and clearly labeled which model corresponds to each dataset.

- Fig. S5A/S6A – style of cartoon used for atom models are inconsistent with the rest of the manuscript and, visually, hard to distinguish

We used the same cartoon style as the rest of the figures, but the bulkier size of the ligands and the multiple binding sites made them visually difficult to distinguish. To improve clarity, we have now redrawn the figures into two panels, ensuring that the ligands and side chains no longer overlap.

Reviewer #4 (Remarks to the Author):

Thank you for your time reviewing our manuscript.

POINT-BY-POINT RESPONSE

Reviewer #1 (Remarks to the Author):

With one exception, the authors have addressed all of my comments satisfactorily. However, I still find the title confusing and suggest changing it to something like “Structural pharmacology of SV2A reveals an allosteric modulation mechanism” or “Cryo-EM structures of SV2A reveal an allosteric modulation mechanism”.

We thank the reviewer for the suggestion. However, our work not only defines the structural basis of SV2A modulation, but also uncovers a principle of allosteric regulation that we believe is generalizable across the superfamily. We feel that this broader framing is valuable to the readership and justifies the inclusion of “major facilitator superfamily” in the title.